# Placental Mesenchymal Stem Cell-Derived Extracellular Vesicles Promote Myelin Regeneration in an Animal Model of Multiple Sclerosis

**DOI:** 10.3390/cells8121497

**Published:** 2019-11-23

**Authors:** Kaitlin Clark, Sheng Zhang, Sylvain Barthe, Priyadarsini Kumar, Christopher Pivetti, Nicole Kreutzberg, Camille Reed, Yan Wang, Zachary Paxton, Diana Farmer, Fuzheng Guo, Aijun Wang

**Affiliations:** 1Department of Surgery, School of Medicine, University of California Davis, Sacramento, CA 95817, USA; kcclark@ucdavis.edu (K.C.); sylvainb972@gmail.com (S.B.); pkumar@ucdavis.edu (P.K.); cdpivetti@ucdavis.edu (C.P.); nbkreutz@email.arizona.edu (N.K.); csreed@ucdavis.edu (C.R.); zjpaxton@ucdavis.edu (Z.P.); dlfarmer@ucdavis.edu (D.F.); 2Shriner’s Hospitals for Children, Northern California, Sacramento, CA 95817, USA; shezhang@ucdavis.edu (S.Z.); yyawang@ucdavis.edu (Y.W.); fzguo@ucdavis.edu (F.G.); 3Department of Biomedical Engineering, University of California Davis, Davis, CA 95616, USA

**Keywords:** mesenchymal stromal cells, extracellular vesicles, multiple sclerosis, myelin regeneration, oligodendrocyte precursor cells

## Abstract

Mesenchymal stem/stromal cells (MSCs) display potent immunomodulatory and regenerative capabilities through the secretion of bioactive factors, such as proteins, cytokines, chemokines as well as the release of extracellular vesicles (EVs). These functional properties of MSCs make them ideal candidates for the treatment of degenerative and inflammatory diseases, including multiple sclerosis (MS). MS is a heterogenous disease that is typically characterized by inflammation, demyelination, gliosis and axonal loss. In the current study, an induced experimental autoimmune encephalomyelitis (EAE) murine model of MS was utilized. At peak disease onset, animals were treated with saline, placenta-derived MSCs (PMSCs), as well as low and high doses of PMSC-EVs. Animals treated with PMSCs and high-dose PMSC-EVs displayed improved motor function outcomes as compared to animals treated with saline. Symptom improvement by PMSCs and PMSC-EVs led to reduced DNA damage in oligodendroglia populations and increased myelination within the spinal cord of treated mice. In vitro data demonstrate that PMSC-EVs promote myelin regeneration by inducing endogenous oligodendrocyte precursor cells to differentiate into mature myelinating oligodendrocytes. These findings support that PMSCs’ mechanism of action is mediated by the secretion of EVs. Therefore, PMSC-derived EVs are a feasible alternative to cellular based therapies for MS, as demonstrated in an animal model of the disease.

## 1. Introduction

Mesenchymal stem/stromal cells (MSCs) are highly proliferative fibroblast-like cells that can be isolated from multiple tissue sources and possess potent regenerative, immunomodulatory, neuroprotective and proangiogenic properties [1]. MSCs have been widely studied to establish reliable and effective cures to numerous diseases, predominantly through the secretion of bioactive factors [2,3,4]. The MSC secretome can include free proteins but also contains extracellular vesicles (EVs), including exosomes, which are nanosized particles that are produced by budding from the endosomal membrane. EVs act as messengers of intercellular communication and can contain bioactive factors, including proteins, lipids and micro-RNAs [5]. EVs have also been proposed as a mechanism by which MSCs provide therapeutic benefits and have immunoregulatory and neuroprotective properties that are achieved through multiple molecular mechanisms [5,6]. Cell-based therapies are limited by potential immune rejection of donor cells and also pose other safety concerns [7]. Increasingly, studies have shown that MSC survival and integration within the host after transplantation are usually poor and that MSCs exert their therapeutic functions mainly via paracrine signaling mechanisms [8]. Therefore, MSC-secreted EVs have promising cell-free-based regenerative therapeutic potential; however, the exact molecular mechanism by which MSCs and secreted-EVs exert immunoregulatory, neuroprotective and proangiogenic properties is poorly understood.

MSC-derived EVs have also been shown to readily cross the blood–brain barrier (BBB) and deliver therapeutic cargo to reduce the effects of neuropathological disease, including multiple sclerosis (MS) [9]. MS is a heterogenous disease that is characterized by demyelination and inflammation caused by immune cell infiltration of the central nervous system (CNS) [10]. MS results from an autoimmune response within the CNS that can lead to neuronal degradation, inflammation and loss of axonal conductivity and gliosis in grey and white matter [11]. Myelin sheaths provide a supportive insulating layer for axons and are produced by oligodendrocytes. Oligodendrocyte progenitor cells (OPCs) are an immature cell subset that can differentiate into mature myelinating oligodendrocytes (OLs) [12]. Loss of oligodendrocytes and accompanying demyelination is associated with progressive axonal degeneration and neurological decline [13]. Current therapies for MS target the immune component of the disease but do not prevent the progressive axonal and neural degradation.

Cellular-based therapies utilizing MSCs are currently being used in clinical trials for the treatment of adult MS [11,14,15]. The placenta has been suggested to be a unique source of MSCs that possess robust immunomodulatory properties and have been reported to be beneficial in murine models of graft versus host disease [16,17]. MSCs derived from the placenta (PMSCs) may be a more appropriate cell source for pediatric diseases because the placenta demonstrates “fetomaternal tolerance” during pregnancy, which is attributed to the expression of human leukocyte antigen-G (HLA-G), a non-classical major histocompatibility complex (MHC) class I molecule that inhibits natural killer cell (NK) killing [18]. Unlike bone marrow-derived MSCs (BM-MSCs), PMSCs express HLA-G on their surface in response to interferon gamma (IFNγ) [19], which is a key inflammatory mediator involved with the onset of MS [10]. Therefore, the expression of HLA-G on PMSCs would make them a unique therapeutic cell source for the treatment of neurodegenerative diseases like MS. Currently, a clinical trial is underway using term PMSCs to treat adult MS and no paradoxical worsening of MS lesion counts has been noted [20].

Our group has previously demonstrated that early-gestational chorionic villus-derived PMSCs can prevent hind limb paralysis in a surgically created lamb model of spina bifida (SB) [21]. The neuroprotective properties of PMSCs occur through paracrine signaling mechanisms. Thorough characterization of PMSC-derived EVs has been performed and it was demonstrated that this unique source of EVs has potent neuroprotective properties and contains key proteins and RNAs that contribute to neuronal survival [22]. The goal of the current study was to determine if PMSC-derived EVs provided regenerative effects in an experimental autoimmune encephalomyelitis (EAE) model of MS and their potential in promoting remyelination. It was hypothesized that PMSC-derived EV treatments would lead to motor function improvement in a dose-dependent manner by preventing oligodendroglia degradation and demyelination associated with MS pathology.

## 2. Materials and Methods

### 2.1. PMSC Isolation and Expansion

PMSCs for this study were previously isolated from the chorionic villus tissue of de-identified discarded second trimester human placenta by explant culture [23]. Early passage (P2-P5) PMSCs were cultured in Dulbecco’s Modified Eagle Medium high glucose (DMEM; GE Life Sciences, Pittsburgh, PA, USA) supplemented with 5% fetal bovine serum (FBS; Atlanta Biologics, Flowery Branch, GA, USA), 20 ng/mL fibroblast growth factor basic (FGF basic; Advent Bio, Elk Grove Village, IL, USA), 20 ng/mL epidermal growth factor (EGF; Advent Bio) and 1% penicillin/streptomycin (P/S; ThermoFisher Scientific, Pittsburgh, PA, USA). Cells were cultured in T150 flasks (Corning Inc., Corning, NY, USA) at 37 °C, under 5% CO_2_ as previously described [23].

### 2.2. PMSC Phenotype

PMSCs were detached using Accutase (Thermo Fisher Scientific) for immunophenotype analysis via flow cytometry. Single suspension PMSCs were first labeled with LIVE/DEAD^®^ Fixable Aqua Dead Cell Stain Kit (Thermo Fischer Scientific) to detect dead cells. PMSCs were then washed and incubated with antibodies directed against CD44 (clone G44-26), CD90 (clone 5E10), CD73 (cloneAD2), CD29 (clone MAR4), CD34 (clone 563), CD31 (clone WM59), and CD45 (clone HI30). Appropriate isotype controls were used for each antibody as previously described [23]. All antibodies were purchased from BD Bioscience (San Jose, CA, USA). All the samples were read by flow cytometry (Attune NxT Flow Cytometer, Thermo Fisher Scientific) and analyzed using FlowJo software (Tree Star Inc., Ashland, OR, USA).

### 2.3. Mixed Leukocyte Reaction (MLR)

To evaluate the immunosuppressive potential of PMSCs, mixed leukocyte reactions (MLRs) were performed exactly as previously described [24]. Briefly, peripheral blood mononuclear cells (PBMCs) were isolated from whole blood by using an underlay method of Ficoll-Paque^TM^ PLUS (GE healthcare, Uppsala, Sweden) and gradient centrifugation. PBMCs were collected from the buffy coat layer and stimulated with 5 µg/mL phytohemagglutin (PHA, Millipore-Sigma, St. Louis, MO, USA). PMSCs were irradiated (10Gy, Varian 2100C linear accelerator, Varian Medical Systems Inc., Palo Alto, CA, USA) and kept on ice before experimental setup. PBMCs and PMSCs were co-cultured at a ratio of 5:1 in DMEM supplemented with 10% heat inactivated FBS, 1% P/S and L-tryptophan (600 μM, Millipore-Sigma).

After 3 days of co-culture, samples were treated with 1 mM Bromodeoxyuridine (BrdU, BD Biosciences). Twenty-four hours following BrdU treatment, leukocytes were collected and cells were stained for LIVE/DEAD^®^ Fixable Aqua Dead Cell to identify live cells and CD3 (clone UCHT1, BD Bioscience) to identify lymphocyte populations. Cells were then stained for nuclear BrdU incorporation per manufacture directions (FITC BrdU Flow Kit, BD Biosciences) and read by flow cytometry (Attune NxT Flow Cytometer, Thermo Fisher Scientific). Flow cytometry data were analyzed using FlowJo software (Tree Star Inc.).

### 2.4. Enzyme-Linked Immunosorbent Assay (ELISA)

PMSCs were cultured for 24 h and supernatants were collected for protein quantification via ELISA as previously described [23]. ELISAs for brain derived neurotrophic factor (BDNF; R&D Systems, Minneapolis, MN, USA), hepatocyte growth factor (HGF; R&D Systems) and vascular endothelial growth factor (VEGF; R&D Systems) were performed per the manufacturer’s instructions and read on a plate reader (SpectraMax Plate Readers, Molecular Devices, San Jose, CA, USA).

### 2.5. Neuroprotection Assay by Indirect Coculture

Neuroprotection assays were performed exactly as previously described [22]. In brief, the SH-SY5Y neuroblastoma cell line was cultured for 24 h. PMSCs were indirectly cultured in hanging well inserts for 24 h. To assess the neuroprotective properties of PMSCs, SH-SY5Y cells were treated with 1 µM staurosporine to induce apoptosis. PMSC inserts were washed and co-cultured with apoptotic SH-SY5Y cells. After 96 h, cells were washed with 2 µM calcein AM (Thermo Fisher Scientific) and imaged using an Axio Observer D1 inverted microscope (Carl Zeiss). Images were processed with WimNeuron Image Analysis (Onimagin Technologies, Cordoba, Spain) for neurite outgrowth analysis.

### 2.6. PMSC-Derived Extracellular Vesicle (EV) Isolation

PMSC-derived EVs were isolated as previously described [22]. In brief, EVs were first depleted from FBS by spinning FBS samples at 112,700 G using the L7 Ultracentrifuge (Beckman Coulter, Brea, CA, USA) and a SW28 rotor for 16 h at 4 °C. Supernatants were collected, aliquoted and stored at −20 °C. PMSCs at passage 4 were seeded at 20,000 cells/cm^2^ in T175 flasks (Corning Inc.) in 20 mL of medium containing 5% EV-depleted FBS, 20 ng/mL FGF (Advent Bio), 20 ng/mL EGF (Advent Bio), and 1% P/S (Thermo Fischer Scientific) for 48 h at 37 °C, under 5% CO_2_. Conditioned medium was collected and EVs were isolated by differential centrifugation exactly as previously described [22]. After the final centrifugation step, EV pellets were resuspended in 10 µL of triple-filtered PBS (GE, Life Sciences) per T175 flask used for the generation of the conditioned medium. EVs were aliquoted and stored at −80 °C.

### 2.7. EV Characterization by Western Blot

To characterize EVs, a Western blotting analysis was performed exactly as previously described [22]. In brief, EVs were treated with NuPAGE LDS Sample Buffer (Thermo Fisher Scientific) and heated to 90 °C. The samples were run, transferred, probed with 1:500 dilution of primary antibodies against Alg-2 interacting protein X (ALIX; rabbit polyclonal, Millipore-Sigma), tumor susceptibility gene (TSG101; clone T5701; Millipore-Sigma), CD9 (clone MM2-57; Millipore-Sigma), calnexin (Clone C5C9, Cell Signaling Technology, Danvers, MA, USA), and CD63 (clone TS63; Thermo Fisher Scientific) in 5% nonfat dry milk in 20 mM Tris-HCl (pH 7.4), 150 mM NaCl, and 0.5% Tween 20 (Millipore-Sigma). Blots were then probed against their respective secondary antibodies and developed using Chemidoc MP Imaging System (Bio-Rad, Hercules, CA, USA).

### 2.8. Characterization of EVs by Nanoparticle Tracking Analysis

Nanoparticle Tracking Analysis (NTA) was performed to quantify EV yield and size distribution. The Nano Sight LM10 Nanoparticle Analysis System (Malvern Panalytical Ltd., Malvern, UK) and the NTA 3.1 Analytical Software (Malvern Panalytical Ltd.) were used to characterize isolated EVs. A 5 µl aliquot of stored EVs was diluted in 995 µl of 0.2 µm triple-filtered Milli-Q water (Millipore-Sigma) and injected into the NTA. A 1:200 dilution is necessary to obtain a concentration between 2 × 10^8^ and 20 × 10^8^ particle/mL, which is the suggested optimal range for NTA [25,26]. Three-30 second videos were recorded and analyzed by the software. The scientific complementary metal-oxide-semiconductor (sCMOS) camera collected 739 frames at 22.0 °C. A 404 nm laser with a detection threshold of 5, determined the mean-square displacement of each nanoparticle based on its own Brownian motion. The analytical software determined EV size, mean, mode and standard deviation, as well as the number of particles per frame and milliliter of sample.

### 2.9. Experimental Autoimmune Encephalomyelitis (EAE) Induction in a Murine Model

In order to mimic the pathology of human MS, EAE was induced using a murine model as previously described [27,28]. Three-month old C57BL/6J mice (male and female) were immunized with myelin oligodendrocyte glycoprotein (MOG) peptide 35–55 to induce EAE (IACUC #19014). In brief, 300 μg of rodent MOG peptide (amino acids 35–55, New England Peptides, Gardner, MA, USA) in Complete Freund’s Adjuvant (CFA) containing 5 mg/mL killed Mycobacterium tuberculosis (Difco, Thermo Fischer Scientific) was administered into the subcutaneous flank of mice at day 0. At day 0, each mouse received two subcutaneous injections of the MOG solution as well as a 100 µl dose of 2 ng/µl pertussis toxins and virulence factors (List Biological Laboratories Inc., Campbell, CA, USA) diluted in sterile PBS (Thermo Fisher Scientific). Pertussis toxins and virulence factors were administered again on day 2. Pertussis toxins allow an increase in the blood–brain barrier permeability in order to facilitate the incursion of the different treatments into the CNS.

In order to monitor disease progression, mice were weighed and scored daily. Neurological deficits were assessed on a five-point scale (limp tail or waddling gait = 1; limp tail and waddling gait = 2; single limb paresis and ataxia = 2.5; double limb paresis = 3; single limb paralysis and paresis of second limb = 3.5; full paralysis of 2 limbs = 4; moribund = 4.5; and death = 5) [27].

### 2.10. Experimental Treatment of EAE Animals

Daily motor quantification was performed. Mouse treatment groups were randomized in order to contain comparable numbers of males and females and an average score close to 3.5 to represent EAE onset. For this study, disease onset and treatment occurred on day 19 post-MOG immunization. On the day of treatment, PMSCs were detached from culture using TrypLE (Thermo Fisher Scientific) and washed twice with PBS (Thermo Fisher Scientific). PMSCs with dimensions of 1 × 10^6^ were resuspended in 200 µl injectable saline and placed on ice prior to administration. Stored EVs were thawed and resuspended at either 1 × 10^7^ (low dose) or 1 × 10^10^ (high dose) EVs in 200 µl of injectable saline.

Tail-vein injections were performed using 1 mL syringes connected to standard hypodermic needles (Covidien, Dublin, Republic of Ireland). EAE mouse scoring was repeated up to day 40 or 43 post-MOG immunization.

### 2.11. Tissue Preparation

Mice were euthanized by CO_2_ asphyxiation and were perfused with ice-cold PBS (Thermo Fisher Scientific). Lumbar spinal cords were harvested, post-fixed in 4% paraformaldehyde (PFA) at room temperature for 2 h, cryopreserved in 30% sucrose overnight, and embedded in OCT. Fourteen micrometer frozen transverse sections were cut on a Leica cryostat.

### 2.12. Quantification of Oligodendroglia Survival

Frozen sections were dried and blocked using PBS (Thermo Fisher Scientific) containing 0.1% Tween 20 (Millipore-Sigma) and 10% donkey serum (Thermo Fisher Scientific) for 1 h at room temperature. Sections were incubated with primary antibodies directed against SOX-10 (clone EP268, Millipore-Sigma) at 4 °C overnight, followed by 2 h incubation at room temperature with secondary antibody. TUNEL staining (Terminal deoxynucleotidyl transferase dUTP nick end labeling) was performed on samples to quantify DNA damaged within SOX10 populations. TUNEL was performed using the In-Situ Cell Death Detection Kit (TMR Red, Millipore-Sigma) per the manufacturer’s instructions. DAPI (Thermo Fisher Scientific) was used to label nuclei, and the sections were mounted with Permount (Thermo Fisher Scientific) for microscopic analysis. Samples were imaged using a Carl Zeiss Axio Observer D1 inverted microscope and analyzed using NIH ImageJ software.

### 2.13. Quantification of Myelin Loss

In order to quantify the loss of myelin in treated EAE mice, Luxol Fast Blue (LFB) was used to stain frozen sections. LFB is a copper phthalocyanine dye that binds to lipoproteins found within the myelin sheath. Frozen sections were dried and rehydrated using 95% EtOH. LFB staining was performed according to the manufacturer’s instructions (IHC World, Woodstock, MD, USA). Sections were mounted with Permount (Thermo Fisher Scientific) and samples were imaged using a Carl Zeiss Axio Observer D1 inverted microscope. Images were analyzed using NIH ImageJ software. Myelin stains blue using LFB, therefore, thresholds were standardized and lack of LFB staining was quantified to denote the percentage of myelin loss in each sample.

### 2.14. EV Functions on Oligodendrocyte Precursor Cell (OPC) Differentiation

Primary OPC culture and differentiation were conducted according to our previous protocol [29]. In brief, OPCs were isolated by immunopanning from neonatal mouse forebrain and maintained in growth medium for population expansion. To study the role of PMSC-EVs in OPC differentiation, 1 × 10^5^ OPCs were switched to the chemically defined differentiation medium with or without PMSC-EVs (2000 × 10^5^). RNA was isolated from differentiating OPCs 24 h after PMSC-EV treatment. RNA isolation was performed using the RNeasy Mini Kit’s manufacturer’s protocol (QIAGEN, Germantown, MD, USA), and RNA concentration was measured with the NanoDrop 2000 Spectrophotometer (Thermo Fisher Scientific). Sybr Green-based RT-qPCR was used to quantify the expression of OL differentiation markers oligodendrocyte-specific molecules (MOG), ectonucleotide pyrophosphatase/phosphodiesterase 6 (Enpp6) and myelin associated glycoprotein (MAG). Data were analyzed using the StepOnePlus Real-Time PCR System (Thermo Fisher Scientific) and processed by the V2.3 StepOne software. For quantification, the mRNA expression level of interested genes in each sample was normalized to the internal control, housekeeping gene *Hsp90* and fold change in gene expression was calculated based on the delta-delta Ct method as previously described [29,30].

### 2.15. Statistical Analyses

The results are expressed as mean and standard error. Imaging and in vitro data were analyzed using non-parametric Mann–Whitney-Wilcoxon t-tests (GraphPad Prism version 8.2.1 for macOS, La Jolla, CA, USA). Multiple comparisons were performed using a Kruskal–Wallis test, followed by Dunn’s post hoc correction to determine which groups were significantly different (GraphPad Prism version 8.2.1 for macOS). A *p*-value < 0.05 was considered significant.

## 3. Results

### 3.1. Treatment of EAE Mice Using EVs and PMSCs

The exact molecular mechanism by which PMSCs confer therapeutic benefits for MS are largely unknown; however, these MSC functions have been shown to occur through paracrine signaling. To determine if PMSCs and secreted factors are suitable for the treatment of MS, an EAE model was utilized. An overview of the experimental design for the current study is summarized in Figure 1A. Induction of EAE was achieved by immunizing C57BL/6J mice with MOG peptide 35–55 and pertussis toxin on day 0 and a secondary injection of pertussis toxin on day 2. Following MOG immunization, disease symptom onset typically peaks at 15–20 days. Motor function scoring was performed on mice daily and peak motor deficiencies in this study were observed on day 19 following MOG immunization. Motor scoring was performed using a previously established scale [27]. Briefly, mice displaying loss of tail tension was scored as a 1, hind limb weakness was denoted as a score of 2 (Figure 1B, panel 1), hind limb paresis was denoted as a score of 3 and dual hind limb paralysis was denoted as a score of 4 (Figure 1B, panel 2). Comparable numbers of males and females were randomized into experimental groups, with each group averaging a score of about 3.5 prior to therapeutic intervention. Experimental treatment groups included sterile saline (negative control), 1 × 10^7^ PMSC-EVs (low dose), 1 × 10^10^ PMSC-EVs (high dose) or 1 × 10^6^ PMSCs (Figure 1C). Notably, many studies investigating the therapeutic utility of MSC-EVs from adult tissues did not provide absolute particle quantification of EV doses. Therefore, a low dose and high dose of EVs were administered to determine if outcomes occur in a dose-dependent manner. Motor scores were performed daily following treatment and mice were euthanized at day 40 or 43 post MOG immunization.

### 3.2. PMSC and PMSC-EV Characterization

In the current study, PMSCs were screened for typical phenotypic markers to identify MSC lineage. PMSCs were positive for the surface expression markers CD73, CD105, CD29, CD90 and CD44 (Figure 2A). Additionally, PMSCs were negative for hematopoietic markers CD31, CD34 and CD45 (Figure 2A). To assess the functional properties of PMSCs, mixed leukocyte reactions were performed to assess the immunosuppressive potential of these cells. PBMCs from multiple donors were stimulated using the mitogen PHA and co-cultured with irradiated PMSCs. PMSCs reduced CD3 positive lymphocyte proliferation of every donor screened (Figure 2B). Additionally, PMSCs secrete high levels of BDNF, HGF and VEGF as measured by ELISA (Figure 2C). Neuroprotection assays were also performed with PMSCs and it was shown that these cells increase total branching points (fold change 2.01), circulatory length (fold change 1.52) and total segment counts (fold change 1.56) in SH-SY5Y apoptotic cells, which aligns with previous study findings [22]. EVs were then isolated from PMSCs that were shown to have both immunomodulatory and neuroprotective properties. Isolated PMSC-EVs express ALIX, CD9, CD63, TSG101 and are negative for calnexin (data not shown). Quantification of EVs for treatment was performed using an NTA analysis, and the average nanoparticle size of PMSC-EVs used in this study was 124.6 +/− 4.1 nm (Figure 2D), which is within the reported size range of EVs (30–150 nm) [26].

### 3.3. PMSC and PMSC-EVs Improve Motor Function Scores in EAE Mice

As compared to saline treated control mice, only high-dose PMSC-EV and PMSC-treated animals showed improved motor functions (Figure 3A). The high dose PMSC-EV treatment group had significantly improved motor function scores as compared to the saline only treatment group (treatment *p*-value = 0.0002). Additionally, PMSCs also significantly improved motor function scores as compared to the saline control treatment group (treatment *p*-value = 0.0002). No significant differences were observed in low-dose PMSC-EV treated animals as compared to saline controls (Figure 3A). Animals were sacrificed at 3 weeks following treatment administration to investigate responses to acute disease onset, mimicking therapeutic intervention during an active MS flare. Interestingly, sex differences in motor function outcomes were noted. In male mice, no changes were observed in low-dose PMSC-EV-treated animals (Figure 3B, panel 1), however, high dose PMSC-EV (Figure 3B, panel 2) and PMSC (Figure 3B, panel 3) treated male mice were different compared to saline controls (treatment effect). However, no statistically significant changes were observed at any given time point following treatment administration (Figure 3B, panels 1–3). Similarly, in female mice, no alterations in low dose PMSC-EV-treated animals was observed (Figure 3C, panel 1). Significant changes in females were only noted in high-dose PMSC-EV (Figure 3C, panel 2) and PMSC (Figure 3C, panel 3) groups, as observed in male mice. However, timelier responses were noted, with female mice showing motor improvement at day 15 following high-dose PMSC-EV (Figure 3C, panel 2) and PMSC treatment (Figure 3C, panel 3). Although sex differences were noted, mixed populations of male and female mice more accurately represent MS patient populations.

### 3.4. PMSCs and PMSC-EVs Protect Oligodendroglia Degeneration in EAE Mice

Motor function data demonstrated that only PMSC and high-dose PMSC-EV-treated animals showed significant improvement and therefore, only these animals were used for the immunohistological analysis. TUNEL (apoptotic cell marker) and SOX10 (oligodendrocyte marker) staining was performed to quantify damaged oligodendrocytes in spinal cord white matter. Representative images of a saline-treated animal (Figure 4A), high-dose PMSC-EV-treated animal (Figure 4B), and a PMSC-treated animal (Figure 4C) are shown. Quantification of SOX10+TUNEL+ cells was performed in both male and female mice. Double positive cells were most abundant in saline-treated animals at lesion sites, therefore, increased magnification of double positive cells is provided in Figure 4D. TUNEL staining in spinal cord tissue sections revealed a decrease in the expression of SOX10+TUNEL+ cells in EAE mice treated with high-dose PMSC-EVs and PMSCs (Figure 4E).

### 3.5. PMSCs and PMSC-EVs Preserve Myelin in the Spinal Cord of EAE Mice

Myelin loss was quantified using Luxol Fast Blue staining, which binds to lipid membranes found within myelin. Percentage of negative staining was quantified within the white matter of the spinal cord. Myelin staining of representative images is shown for saline (Figure 5A), high-dose PMSC-EV (Figure 5B) and PMSC (Figure 5C) treated animals. Demyelination occurred most often within spinal cord lesions, notably in saline-treated animals. Myelin staining was greatly reduced in saline mice at lesions sites. These areas are highlighted by arrow indicators in Figure 5A. The quantification strategy for demyelination is shown in a saline-treated mouse lesion (Figure 5D). Magnification of the lesion site within the white matter is shown in panel 1 of Figure 5D. Images were converted to black and white (Figure 5D, panel 2) and colors were inverted to highlight negative staining patterns (Figure 5D, panel 3). A standardized thresholding strategy was implemented and negative black pixel intensity was quantified as a percent myelin loss in white matter (Figure 5D, panel 4). Compared to saline treated EAE mice, high-dose PMSC-EV and PMSC-treated animals showed a significant reduction of myelin present in the spinal cord (Figure 5D). No observable alterations were noted between high-dose PMSC-EV and PMSC-treated animals.

### 3.6. PMSC-EVs Promote OPC Differentiation In Vitro

To investigate whether PMSCs preserve oligodendrocyte populations or drive OPC differentiation to OLs, an OPC differentiation assay was performed with PMSC-derived EVs. Primary OPCs isolated from neonate mice were treated with PMSC-EVs and oligodendrocyte differentiation markers MOG, Enpp6 and MAG were quantified using RT-PCR. Gene expression was compared between untreated OPCs and PMSC-EV treated OPCs. Both MOG (Figure 6A) and Enpp6 (Figure 6B) were significantly upregulated in the presence of PMSC-EVs. There was also a trend of increased expression of MAG (Figure 6C), however, this was not statistically significant.

## 4. Discussion

The goal of the present study was to investigate the use of a novel early-gestational chorionic villus-derived MSC source for the treatment of MS. Furthermore, we aimed to investigate whether PMSC-derived EVs confer therapeutic benefits and whether these particles can be used as a cell-free treatment for neurodegenerative disease. Comparable to findings from other studies using adult-derived MSCs [11], PMSCs achieved motor improvement in an EAE murine model of MS. Improvements in motor function were achieved in part through a neuroprotective mechanism. PMSC-treated animals displayed less DNA damage within oligodendrocyte populations and myelin was preserved in the spinal cords of these animals. Interestingly, in the treatment group, animals that did not show motor improvements, mild increases of DNA damage and demyelination were noted. Additionally, similar protective effects preserving oligodendroglia and myelination were also achieved by treatment with high-dose PMSC-derived EVs. These effects were not noted in the low dose PMSC-EV-treatment groups, suggesting that the PMSC secretory mechanism of action occurs through EV signaling in a dose-dependent manner.

These findings demonstrate that the clinical benefits of PMSCs can be achieved by treatment with PMSC-EVs alone. Our group has previously demonstrated that PMSCs secrete high levels of BDNF, HGF and VEGF as compared to BM-MSCs [21]. Proteomic analysis of PMSC-EVs revealed the presence of HGF and VEGF in these nanoparticles. HGF, in particular, has been shown to be secreted in MSC-conditioned medium and can mediate motor recovery functions in an EAE model of MS [30]. HGF is a potent angiogenic factor that has been shown to exert immunomodulatory effects through the stimulation of regulatory T cells, which, in turn, mediate autoimmune responses [31]. These factors may play a role in the protective and regenerative properties of PMSCs. In this study, high-dose PMSC-EV treatments lead to comparable responses to PMSC treatments. This suggests that PMSC-mediated clinical improvements in the current EAE model occurs through an EV-meditated mechanism. The presence of these factors in PMSC-EVs also suggests that these mediators may play an important role in protective, regenerative and immunomodulatory properties of PMSCs. Additionally, PMSCs secrete higher levels of these proteins compared to BM-MSCs [21], thus, this unique cell source may result in improved clinical outcomes compared to adult-derived MSC sources.

Our in vivo data demonstrate increased myelin present in spinal cord white matter of PMSC and PMSC-EV-treated EAE mice. In these animals, oligodendrocyte survival was increased as compared to saline-treated controls. PMSC and high-dose PMSC-EV treatments led to less DNA damage to oligodendroglia. Oligodendrocytes are myelin producing cells; therefore, increased levels of oligodendrocytes correspond to increased myelin levels. To elucidate whether PMSC-EVs promote oligodendrocyte maturation, differentiation assays were performed. In vitro data demonstrate that PMSC-EVs drive OPCs to express OL differentiation markers, which suggests that PMSC-EVs are promoting maturation of myelinating oligodendroglia. These findings suggest that PMSCs and PMSC-EVs have both protective and regenerative properties, exhibited by driving OPC differentiation. Additionally, interactions of oligodendrocytes and autoreactive T lymphocytes are not well understood, therefore, investigations of these interactions in response to PMSC and PMSC-EV therapy are warranted.

Limitations of using an EAE model have been widely noted, and although the model is reliable to induce MS-like symptoms, including inflammation and demyelination, variation in disease onset occurs even within genetically identical rodents [32]. Variability in disease onset was observed in this study; however, a standardized average score within each experimental group was used prior to experimental treatment to account for these alterations. This variability likely affected motor function and recovery data and could explain why some animals did not respond to treatment with PMSCs or high-dose PMSC-EVs. Additionally, for the purposes of this study, both male and female mice were used; however, a timelier improvement was noted in female mice. Sex differences have been noted in EAE onset, which reflects higher incidences of MS in human female populations [33]. These alterations may be due to effects of androgens on T lymphocytes, which are key mediators in the pathologic features of MS [34]. Sex differences in response to EAE onset methodologies can occur in multiple mouse strains, and it has been shown that male mice do not result in robust Th1 responses and instead shift to anti-inflammatory Th2 responses as compared to females [33]. Sex differences are not typically noted by C57BL/6 mice; however, in the current study, EAE onset by males and females differed, but to recapitulate MS patient populations, both sexes were included for analysis.

Myelin degeneration is a main pathologic feature of MS; however, autoimmune infiltration of Th1 pro-inflammatory cells into the CNS is a key contributor to the inflammatory and degenerative properties of MS [35]. T regulatory cells modulate Th1 and Th2 responses and have impaired functions in MS patients [36]. MSCs have been shown to induce T regulatory populations and drive shifts of Th1 to Th2 responses [36]. The induction of T regulatory responses by MSCs is a mechanism by which these cells modulate immune responses and may have clinical utility for autoimmune diseases such as MS. Given that male mice have a stronger tendency towards Th1 responses in EAE [33], this could explain the modest motor recovery responses observed in these animals to PMSC and PMSC-EV treatment. While the focus of the present study was on the myelin protective and regenerative properties of PMSCs and PMSC-EVs, future studies will investigate the immunoregulatory properties of these cells and nanovesicles within immune cell subsets.

Although PMSC and PMSC-EV treatment resulted in motor improvements following a single injection in mice in the acute phase of EAE, future studies are needed to address dosage timing and strategy. For this study, small volumes of diluted EVs and PMSCs were injected via tail veins of mice. However, it has been reported that PMSCs have poor homing and engraftment [2]. Additionally, nanoparticle treatments pose technical challenges because injections need to be precise to ensure proper and complete administration of therapeutic doses. In future studies, tracking cells and EVs, as well as evaluation of different administration modalities will be performed. Biodistribution studies will provide key insights to further improve clinical outcomes for MS patients.

The findings from this study demonstrate that PMSC-EVs are a viable option for the treatment of neurodegenerative diseases and pose several advantages compared to cellular-based therapies. EVs provide many benefits, including immunotolerance, storage stability, heterogenous cargo and multiple therapeutic outcomes. This study provides key preliminary data that will facilitate future studies investigating the use of PMSC-EVs for the treatment of MS.

## 5. Conclusions

In an EAE rodent model of MS, PMSCs exert therapeutic benefits in part by preserving or driving oligodendroglia differentiation and myelination. Our data demonstrate that PMSC-EVs achieve similar clinical outcomes as PMSC treatments in a dose-dependent manner. The findings from this study provide evidence that PMSCs are a unique cell source for the treatment of neurodegenerative diseases and that therapeutic benefits can be achieved using isolated EVs. Additionally, PMSCs may have superior clinical benefits in the context of pediatric neurodegenerative diseases. This study will be used as a platform for further investigation of the molecular mechanism and intracellular communication between cell subsets to improve potential treatment strategies for MS.

## Figures and Tables

**Figure 1 cells-08-01497-f001:**
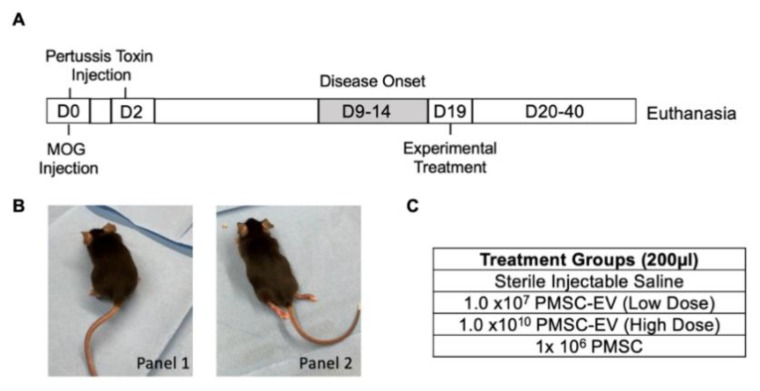
Experimental design overview. (**A**) Three-month old C57BL/6J mice (male and female) were immunized with myelin oligodendrocyte glycoprotein (MOG) peptide 35–55 to induce EAE on Day 0. MOG peptide in Complete Freud’s Adjuvant (CFA), and pertussis toxin was injected into mice on day 0. Pertussis toxin was administered again on day 2 following MOG immunization. Disease onset in EAE peaks 15–20 days following MOG administration. For this study, peak disease onset occurred on day 19, which was determined using motor function scores. Treatments were administered on day 19 and motor function scores were recorded for 40–43 days following MOG immunization, after which animals were sacrificed for further analysis. (**B**) Animals were scored on a 5-point scale, and groups were randomly assigned, averaging a score of 3.5. Animals with mild hind limb deficits but still weight bearing were marked as a score of 2 (panel 1) and animals with complete hind limb paralysis were marked as a score of 4 (panel 2). (**C**) Treatment groups included saline (negative control), low-dose placenta-derived MSC (PMSC)-derived extracellular vesicles (EVs0, high-dose PMSC-derived EVs and placenta-derived MSCs (PMSCs).

**Figure 2 cells-08-01497-f002:**
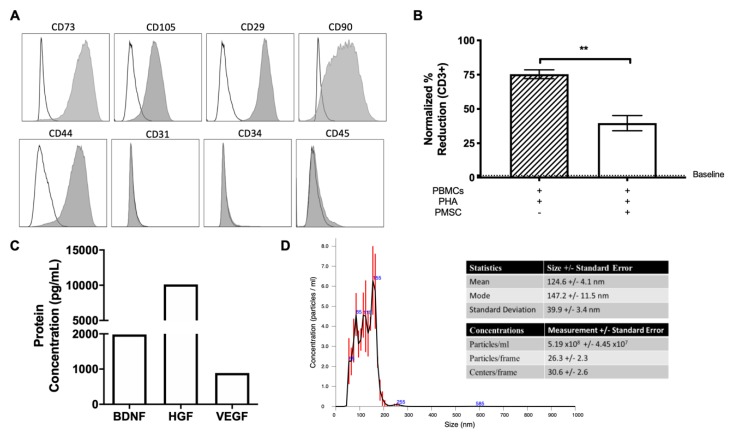
PMSC and PMSC-EV characterization. (**A**) PMSCs were analyzed using flow cytometry and were positive for typical MSC markers CD73, CD105, CD29, CD90 and CD44. Additionally, PMSCs were negative for the hematopoietic markers CD31, CD34 and CD45. Data are presented as median fluorescence intensity (MFI) overlays to respective negative isotypes. (**B**) Mixed lymphocyte reactions were performed to assess the immunosuppressive potential of PMSCs. Peripheral blood mononuclear cells (PBMCs) from multiple donors (*n* = 3) were stimulated with the mitogen Phytohemagglutinin (PHA) and co-cultured with irradiated PMSCs. Suppression of leukocyte proliferation was measured as a percentage of BrdU incorporation within CD3-positive PBMCs using flow cytometry. Unstimulated PBMC proliferation is denoted by a dashed line. (**C**) PMSCs secrete high levels of brain-derived neurotrophic factor (BDNF), hepatocyte growth factor (HGF) and vascular endothelial growth factor (VEGF). (**D**) Extracellular vesicles (EVs) were collected from passage 4 PMSCs and used for further characterization analysis. Nanoparticle tracking analysis (NTA) measurements determined the size and concentration of PMSC-derived EVs. EV size had a mean of 124.6 +/− 4.1 nm. Data are represented as mean and standard error. * *p* < 0.05, ** *p* < 0.01.

**Figure 3 cells-08-01497-f003:**
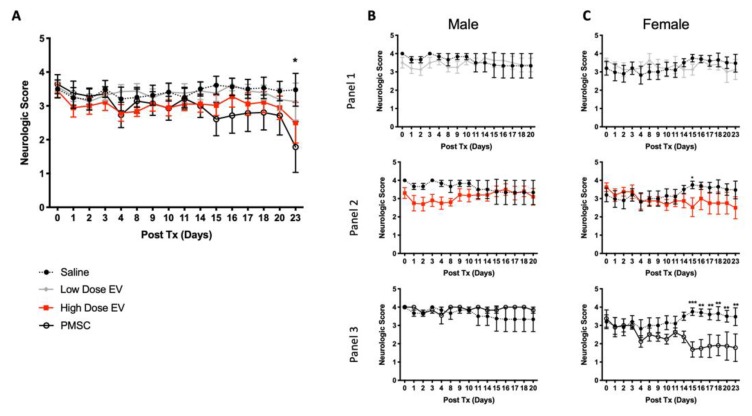
PMSC and high-dose PMSCs-EV treatment improves motor function in EAE mice. Animals were treated at peak disease onset following MOG immunization. (**A**) PMSC and high-dose PMSC-EVs significantly improve motor function as compared to saline treated animals (treatment effect). Compared to baseline-only, high-dose PMSC-EV and PMSC-treated animals show motor improvement at day 23 following treatment administration. No observable changes were noted in low-dose PMSC-EV treated animals. Interestingly, altering responses were noted in male and female mice and further analysis was performed on each sex. (**B**) Male mice displayed no significant alterations in the low dose PMSC-EV treated group (panel 1) as compared to saline controls; however, both high-dose PMSC-EV (panel 2) and PMSC (panel 3) treated animals had significant responses to treatment. However, these findings did not reveal timewise alterations. (**C**) Female mice similarly displayed no changes in the low-dose PMSC-EV treatment group (panel 1) compared to saline controls. Females did, however, display significant improvements in motor function in high-dose PMSC-EV (panel 2) and PMSC (panel 3) treatment groups. Additionally, these improvements occur in a timelier manner beginning at day 15 following high dose PMSC-EV and PMSC administration as compared to saline mice. Data are presented as mean and standard error. * *p* < 0.05, ** *p* < 0.01, *** *p* < 0.001.

**Figure 4 cells-08-01497-f004:**
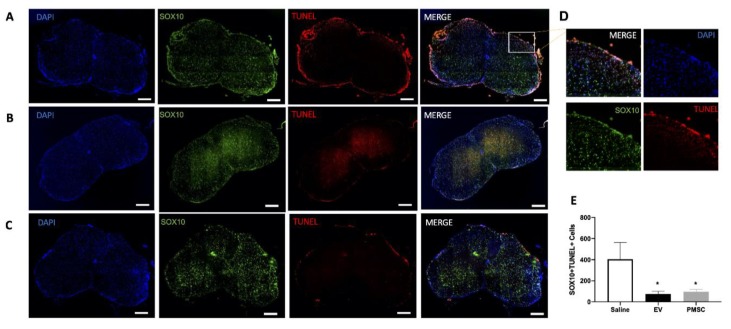
PMSC and PMSC-EV treatment reduce oligodendrocyte damage in EAE mice. Representative images of spinal cords from EAE mice treated with (**A**) saline (control), (**B**) high-dose PMSC-EVs or (**C**) PMSCs are shown at 20× magnification. Spinal cord sections were stained with SOX10 (green) to denote oligodendrocytes. TUNEL staining was also performed to identify DNA damage within oligodendrocyte populations. (**D**) Increased numbers of SOX10+TUNEL+ cells were observed in lesion sites within the white matter, most often in saline-treated animals. Increased magnification of a lesion from a saline treated mouse is shown. (**E**) Compared to saline-treated animals, high-dose PMSC-EV and PMSC-treated animals had a significant reduction of SOX10+TUNEL+ positive cells. Scale bar = 200 μM. * *p* < 0.05.

**Figure 5 cells-08-01497-f005:**
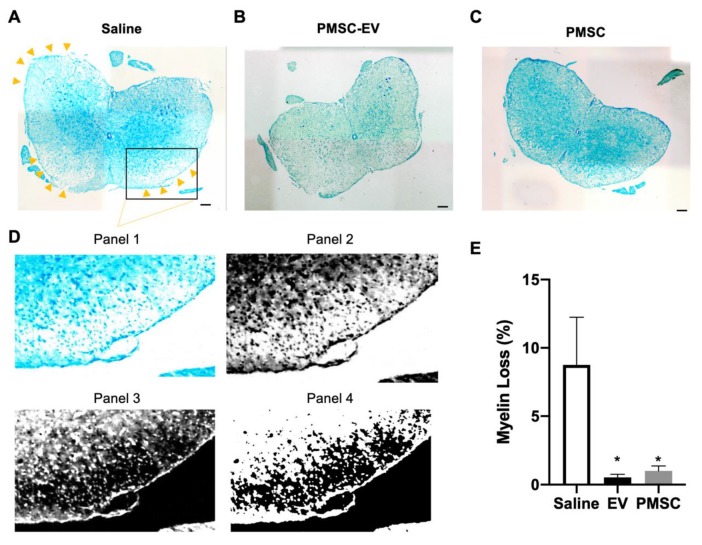
PMSC and PMSC-EV-treated EAE mice show decreased myelin loss. Spinal cords from EAE mice treated with saline (control), high-dose PMSC-EVs or PMSCs were stained with Luxol Fast Blue, a dye that binds to myelin. Representative images of (**A**) saline, (**B**) high-dose PMSC-EV and (**C**) PMSC-treated animals are shown at 10x magnification. (**D**) Magnification of a demyelinating lesion site in the saline control animal is highlighted in panel 1. Quantification of demyelination was performed by converting images to black and white (panel 2) and inverting colors, resulting in demyelinating areas shown in black (panel 3). Standardized thresholding was applied (panel 4) and quantification was performed using pixel intensity of negatively stained areas in white matter. (**E**) Myelin loss within the white matter of mouse spinal cords was recorded. Compared to saline-treated animals, high-dose PMSC-EV and PMSC treated animals showed a significant reduction of myelin loss. Scale bar = 200 μM. * *p* < 0.05.

**Figure 6 cells-08-01497-f006:**
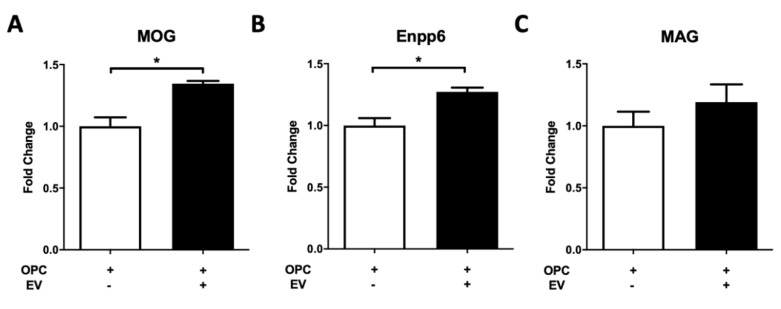
PMSC-EVs drive oligodendrocyte precursor cells (OPCs) to a mature lineage phenotype. PMSC-EVs were added to primary murine OPC cultures. Expressions of mature oligodendrocyte markers myelin oligodendrocyte protein (MOG), ectonucleotide pyrophosphatase/phosphodiesterase 6 (Enpp6) and myelin associated glycoprotein (MAG) were evaluated. PMSC-EVs promoted significant upregulation of MOG (**A**) and Enpp6 (**B**). Although not statically significantly, a trend in increased expression of MAG (**C**) was noted. Data are represented as mean and standard error. * *p* < 0.05.

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
