# Peer review of "Placental Mesenchymal Stem Cell-Derived Extracellular Vesicles Promote Myelin Regeneration in an Animal Model of Multiple Sclerosis"

_cells, 2019, doi:10.3390/cells8121497_

Round 1
Reviewer 1 Report
In this work the authors sought to study the effects of both PMSCs and PMSC-derived EVs in an experimental mouse model of MS. This is a very well written mns with a very good study design. The authors noted a dose-dependent effect of EVs which is very novel.Â
Comments/edits:
Page 1, Lines 39, 40: "and can contain contents including.." did the authors meant "can contain biactive factors including.."
Page 2, line 89: "cultured in medium containing Dulbecco’s Modified Eagle Medium high glucose"Â
Page 2-3, Lines 93-94: Cells were cultured in T150 flasks (Corning Inc., Corning, NY, USA) and cultured at 37 ÌŠC, under 5% CO2
Page 3, lines 110-111: PBMCs were collected from the buffy coat layer an were collected for co-incubations. PBMCs were and stimulated....
Page 3, line 114: in medium containing DMEM supplemented...
Page 4, line 159: 2.8. EV Quantification Characterization of EVs by Nanoparticle Tracking Analysis
Page 4, line 180: Pertussis toxins and virulence factors was were
Page 5, line 191: TypLE TrypLE: Is there a specific reason the authors use Accutase before (page 3, line 97) and then use TrypLE?
Page 7, lines 293, 294: it was shown that the that the
Page 7, line 299: Please define MFI
Page 8, line 305: n=?
Figure 2D: Axis labels are blurry. Text in table is to small
Page 8, line 319: intervention during and active MS flare
Page 8, line 323: treated male mice were significantly different...
Page 10, line 397: Primary OPCs were isolated from neonate mice and then were treated with PMSC-EVs. and the The...
Results:
The results of the neuroprotection assays are not shown in any figure, but mentioned in the text. Neuroprotection assays were performed with PMSCs. Did the authors perform these assays using EVs? this may be a good comparative experiment to show.
Discussion:Â
Please elaborate more regarding the sex differential effect, this is a key finding and understanding the this differential effect could be important. The authors show that PMSCs secrete large quantities of BDNF, HGF and VEGF. Please discuss further on the importance of this finding in the context of MS and the model in study. Have EVs been analyzed in terms of the contents of these factors? Motor improvement functions is higher with PMSCs compared to EVs (figure 3C). EVs effect is better in reducing oligodendrocyte damage and myelin loss. Could the authors discuss further?
Â
Reviewer 2 Report
Thank you for submitting this study.Â
Addressing repair and regeneration in MS is absolutely essential. MSCs hold some promise in this regard. Adding EVs is an intriguing step froward which could be interesting.Â
You have elected to test these techniques at the peak of EAE expression. The results obtained for motor function are modest, the average score improving from 3,5 to 2,5. This could indicate that reconstituting myelin and improving oligo survival are insufficient in themselves. In figure 3, panel 3C the mice without the high dose EVs, fare better than with EVs.Â
Do you have data showing increased maturation of OPCs in vivo?Â
As MS is relapsing and remitting, it would be interesting to repeat the experiments earlier in the course, which could improve results. Â
